# An Animal Model for Assessing the Effects of Hydroxyurea Exposure Suggests That the Administration of This Agent to Pregnant Women and Young Infants May Not Be as Safe as We Thought

**DOI:** 10.3390/ijms19123986

**Published:** 2018-12-11

**Authors:** Lucía Rodríguez-Vázquez, Joaquín Martí

**Affiliations:** Unidad de Citología e Histología, Facultad de Biociencias, Universidad Autónoma de Barcelona, Bellaterra, 08193 Barcelona, Spain; lucia.rodriguez.vazquez.ae@gmail.com

**Keywords:** hydroxyurea, cerebellum, neuron, immunohistochemistry, electron microscopy, cell death, apoptosis

## Abstract

The cytostatic agent hydroxyurea (HU) has proven to be beneficial for a variety of conditions in the disciplines of oncology, hematology, infectious disease and dermatology. It disrupts the S phase of the cell cycle by inhibiting the ribonucleotide reductase enzyme, thus blocking the transformation of ribonucleotides into deoxyribonucleotides, a rate limiting step in DNA synthesis. HU is listed as an essential medicine by the World Health Organization. Several studies have indicated that HU is well tolerated and safe in pregnant women and very young pediatric patients. To our knowledge, only a few controlled studies on the adverse effects of HU therapy have been done in humans. Despite this, the prevalence of central nervous system abnormalities, including ischemic lesions and stenosis have been reported. This review will summarize and present the effects of HU exposure on the prenatal and perinatal development of the rat cerebellar cortex and deep cerebellar nuclei neurons. Our results call for the necessity to better understand HU effects and define the administration of this drug to gestating women and young pediatric patients.

## 1. Introduction

Hydroxyurea (HU) is an inhibitor of the ribonucleotide reductase enzyme. It impairs DNA synthesis in a wide variety of cells and organisms, including *Saccharomyces cerevisiae* [1]. This antimetabolite has provided therapeutic benefits in the treatment of neoplastic diseases and hemopathies, and it has emerged as an important option for many pregnant women and pediatric patients with sickle cell anemia [2]. Estimates indicate that approximately 250,000 children are born annually with this hematological disorder worldwide [3]. Several case reports suggest that HU may have minimal or no major adverse effects on the development of the human fetus and in very young children [4,5]. However, the Center for the Evaluation of Risks to Human Reproduction is concerned that HU may increase the risk for congenital anomalies or developmental abnormalities in fetuses after exposure to pregnant women [6]. Moreover, anemia and central nervous system abnormalities have been reported in young pediatric patients treated with HU [7].

Previous research has revealed that the administration of this teratogenic agent to embryo rodents generates the apoptosis of neuroepithelial cells in the fetal telencephalon [8,9]. In perinatal life on the other hand, cerebellar external granule cell (EGL) depletion and the ectopic location of granule cells (GCs) due to the HU exposure have been reported [10,11]. Despite these results, the influence of this agent on the development of the rat cerebellum has not been completely elucidated. The cellular response to damage induced by HU has received little attention. We will show here that the results of our research have important implications for the administration of HU in pregnant mothers and infants.

This review is focused on the development of the rat cerebellum following treatment with a single dose of the cytotoxic agent HU. Two procedures were followed, in the first one, rats were exposed to HU in utero and sacrificed at regular intervals from 5 to 35 h after drug administration or later once they had reached adulthood. In the second procedure, animals were treated with HU during their perinatal stage of life and killed at appropriate times, ranging from 6 to 48 h after treatment administration or later once they had reached adulthood.

## 2. Hydroxyurea: An Overview

HU is water soluble, has a low molecular weight and is a non-alkylating compound (chemical formula CH_4_N_2_O_2_) [12] that was first synthesized in Germany almost 150 years ago by Dresler and Stein in a series of experiments attempting to extract derivatives from urea [13]. At first, in 1928, it was observed that HU induced leukopenia, anemia macrocytosis and death. In the late 1950s, this drug underwent further preclinical testing and was noted to have a significant effect against LI210 leukemia cells and various solid tumors [14]. The clinical use of HU as an anti-tumor agent began in the 1960s [2,15]. The therapeutic spectrum of this medicine has been expanded for the treatment of patients with chronic myeloid leukemia, essential thrombocytosis, polycythaemia vera and sickle cell anemia [12,16]. HU is also used for the management of dermatological conditions, including psoriasis [17] and HIV infection [18]. HU is currently listed as an “essential medicine” by the World Health Organization [19]. 

## 3. Mechanisms of Action of the Hydroxyurea

The effect of the HU is cell-cycle specific. The drug acts in the S phase, causing an arrest of proliferating cell populations in the G_1_/S phase of the cell cycle [12,14]. To this day, the mechanism of action of HU remains incompletely understood [20]. Previous studies have indicated that this agent cleaves the DNA molecule directly [21]. On the other hand, Weinlich and Fritsch [22] showed that HU alters thymidine incorporation during DNA replication, which results in the inhibition of this process, thereby impairing cell proliferation. Moreover, this agent affects the DNA by fragmenting the metaphase chromosomes [23]. An alternative mechanism that has been proposed suggests that HU may kill the cells via the generation of oxidative stress [24].

The ribonucleoside reductase is a multisubunit enzyme responsible for the reduction of ribonucleotides to their corresponding deoxyribonucleotides, which are the building blocks for DNA replication [25]. HU inactivates this enzyme by quenching the tyrosyl free radical required for enzyme activity and a spontaneous regeneration of the active enzyme occurs when the HU is removed [25,26]. Inhibition of the class I form of ribonucleotide reductase blocks the transformation of ribonucleotides into deoxyribonucleotides, depleting the intracellular deoxynucleotide triphosphate pool and halting DNA synthesis. This mechanism of action causes DNA replication fork stalling and leads to the formation of strand breaks [27], all without interfering with the synthesis of ribonucleic acids or proteins [2,15,16,17,18,19,20,21,22,23,24,25].

## 4. Teratogenic Effects of Hydroxyurea

Clinical experience with HU has been accumulated for the past 25 years. The bulk of the current evidence suggests that this antimetabolite is a well-tolerated medication and is efficacious for many gestating women and pediatric patients with hematological diseases [28]. Several studies have revealed that HU may have minimal to no effect on developing human fetuses [29,30] and very young children [4]. The dosage of HU employed in cancer therapy has varied from 40 to 80 mg/kg and is typically administered only once or repeated times. In non-neoplastic diseases, the doses have varied from 15 to 30 mg/kg per day [12]. During pregnancy, the amount has ranged from 0.5 to 6 g/kg per day [29]. However, the Center for the Evaluation of Risks to Human Reproduction of the National Institute of Environmental Health Sciences has indicated that HU exposure during pregnancy may have adverse effects on fetuses [6,31], as anemia and central nervous system abnormalities (ischemic lesions and stenoses) have been reported in pediatric patients [7]. 

There are many reports indicating that HU is a potent mammalian teratogenic agent. When administered to pregnant dams, it induces in the offspring a myriad of effects, such as the loss of mesenchymal cells in the lungs [32], alterations in the craniofacial tissues [33], and malformations in the hindlimbs, tail and neural tube, which seem to be mediated by the activation of p38 mitogen-activated protein kinase pathways [34]. HU also causes microencephaly, hydrocephalus [35] and apoptosis, mediated by the tumor protein p53, located in the neuroepithelial cells of the mouse fetal telencephalon [8,9]. On the other hand, rodents exposed to this hydroxylated derivative of urea during perinatal life present cell depletion in the cerebellar EGL and ectopic location of GCs [10,11].

## 5. Justifying the Choice of the Cerebellum as a Model to Assess the Effects of Hydroxyurea Exposure

The rat cerebellum has been chosen as a model to analyze and interpret the toxic effects of HU exposure to the development of the central nervous system for two reasons: 

(1) The organization of neuron populations in the cerebellar cortex and the uniformity of synaptic circuits make this area a useful model to study the cellular and molecular mechanisms underlying neuronal development [36]. The analysis of the cerebellar development can serve as a system model to investigate the effects of insulting agents that alter the normal pattern of central nervous system development. This metencephalic region is composed of a limited number of neuronal phenotypes that are specifically integrated in a corticonuclear network and are characterized by a distinctive morphology and molecular markers [37,38,39]. Many lines of evidence have indicated that cerebellar neurons are produced following strict neurogenetic timetables [36]. Moreover, neuron generation is compartmentalized, with ventricular zone progenitors giving rise to gamma-aminobutyric acid-ergic (GABAergic) neurons and the rhombic lip precursors to glutamatergic cells [40,41,42]. During cerebellar development, GC precursors also arise from the rhombic lip, but they migrate tangentially to form a secondary matrix, the EGL [36].

(2) The cerebellum is highly vulnerable to insults, the expression of mutant genes and intoxication [43,44,45]. In this context, it has been reported that several spontaneous murine mutations, including lurcher, staggerer, reeler and weaver cause a marked reduction in the volume and shape of the cerebellum due to the severe depletion of neurons [46,47,48]. In addition, it has also been indicated that the selective elimination of neuron precursors with X-irradiation [36], platinum compounds [49,50], impairment of the thyroid status [51], ethanol [52] and virus infection [53] produce an important reduction in size of the cerebellar cortex.

## 6. Embryonic Effects of HU Exposure: Short-Survival Experiments

The development of the cerebellum consists of a series of sequential morphogenetic transformations that begin with the proliferation of the neuronal precursors and end with the myelination of axons of the fiber tracts. During the development of the cerebellar primordium, a specialized germinal matrix, the neuroepithelium, gives rise to several types of neurons, including Purkinje cells (PCs), interneurons and deep cerebellar nuclei (DCN) neurons [54]. Previous research has indicated that the administration of HU to pregnant mice produced the apoptosis of neuroepithelial cells in the fetal telencephalon [8,9]. As the cerebellar neuroepithelium presents a cohort of asynchronous cycling cells in S phase, neurotoxic factors that disrupt the proliferative activity of neural progenitors, such as HU, may cause cell death in the cerebellum. Here, we present an immunohistochemical and ultrastructural study of the cell death of cerebellar neuroblasts following treatment with the fetotoxic compound HU. The data in this section are taken from our previously published results [55].

Pregnant rats were treated with a single dose of saline or HU (600 mg/kg i.p.) on embryonic days (E)13, 14 or 15 and their progeny was sacrificed at regular intervals from 5 to 35 h after drug administration. The quantification of several parameters such as the density of pyknotic, mitotic and PCNA-reactive cells denotes that the administration of HU disrupts the proliferative behavior of neural progenitors in the cerebellar neuroepithelium and induces deleterious effects on this structure. Despite that, we have observed that, for reasons that have still not been elucidated, some neuroepithelial cell precursors have the capacity to resume mitotic activity after being injured. These observations indicate that the degree of damage induced by HU may influence the level of abnormalities in the cerebellum. As PCs and DCN neurons are the first cells that arise from the neuroepithelium [36,41,42] and the time windows of HU exposure (from E13 to E15) correspond with the developmental timetables of these neurons [36,56], we propose that several PCs and DCN neurons may have never been produced.

Apoptosis is a specific type of cell death, a natural process that occurs during morphogenesis of the nervous system. Ultrastructural features of apoptosis involve condensation and margination of the chromatin, cell shrinkage, membrane blebbing and nuclear fragmentation [57]. In order to determine whether the HU administration triggers apoptotic cell events in the cerebellar neuroepithelium, and if so, which type of cell death, TUNEL staining and transmission electron microscopy were used. Our microscopic observations will demonstrate that exposure of the embryonic cerebellum to HU induces apoptotic cell death in a large number of neuroblasts.

In the neuroepithelium, few apoptotic cells can be found during the normal development of the cerebellum. However, a progressive increase in the density of TUNEL-reactive cells was observed between 5–30 h after HU administration (Figure 1a). The maximum density was found at 30 h. After that, values declined.

At all time points, electron microscopy showed the presence of dying cells scattered throughout the neuroepithelium at different stages of apoptosis (Figure 1b–f). The earliest morphological signs of this type of cell death were observed in the nucleus. Chromatin condensation and segregation at the nuclear periphery were a typical initial characteristic. In some instances, chromatin compaction was associated with the convolution of the nuclear envelope, giving a star-like appearance. The mid-to-late apoptotic cell was characterized by the presence of several round and electron-dense nuclear fragments, which were surrounded by cytoplasm but apparently were devoid of membranes. Following HU treatment, numerous clusters of apoptotic bodies were visualized throughout the neuroepithelium, which exhibited an acute level of cytoplasmic degradation and fragmentation. Other ultrastructural changes included apoptotic cells engulfed by surrounding phagocytic cells and dying cells, with unusual apoptotic features, such as a vacuolated cytoplasm, deteriorated organelles and a recognizable nucleus.

Two major points can be deduced from these results: The administration of HU induces cell death by apoptosis on the cerebellar neuroepithelium, and secondly that the use of HU would be a good model for studying the basic histological and ultrastructural features of cell apoptosis. 

## 7. Embryonic Effects of HU Exposure: Long-Survival Experiments

The adult cerebellum appears to be similarly organized across mammals. The cerebellar cortex is formed by three layers: The molecular layer (ML), whose neuronal components include stellate and basket neurons; the Purkinje cell layer, that contains PCs and candelabrum cells; and the granule layer, that consists of GCs, Golgi cells, unipolar brush cells and Lugaro cells [58,59]. The neural phenotypes that populate the cerebellum are generated following regular and precise timetables of neurogenesis [36]. Moreover, previous studies have demonstrated that, in normal rodents, cerebellar system neurons are distributed following precise neurogenetic gradients, i.e., early-produced neurons settle in different locations than late-born neurons [36]. Here, we show that a single administration of HU during embryonic life modifies the regular cytoarchitecture of the cerebellum and alters the neurogenetic profiles and settled patterns of PCs and DNC neurons. The results of this section are taken from a previous published paper [48].

The pregnant dams were administered with a single injection of saline or HU (600 mg/kg i.p.) on day E12. After this treatment, they were injected several times with 6 mg of 5-bromo-2’-deoxyuridine (BrdU) in accordance with the procedure of Sekerkova et al. [60]. This marker was delivered following a progressively delayed labeling comprehensive procedure [36,56] that consisted of injecting pregnant dams in an overlapping series, in accordance with the following time-windows: E13–14, E14–15, E15–16 … E19–20. The offspring were sacrificed at postnatal day 90 (P90) and several cerebellar features were quantified, per section, in each cerebellar cortex compartment (vermis, paravermis, and medial and lateral hemisphere), or alternatively in each deep nucleus (fastigial, interposed and dentate): The area of the cerebellum, length of the cerebellar cortex, ML area, PC number, GC number, area of the IGL, area of the white matter, the areas of the cerebellar nuclei, and number of DCN neurons.

Our results revealed no signs of toxicity in the pregnant dams after HU treatment. They gave birth as normal. Moreover, no sex differences were observed in the effect of the drug when the males and females were compared. Our data also indicated that HU exposure does not compromise neither the cytoarchitecture of the cerebellar cortex nor the deep nuclei. However, it was observed that HU administration contributed to an important cerebellum size reduction. This deficient growth occurred in each analyzed cerebellar compartment and in deep nuclei. These results suggest that the effect of the HU exposure is toxicologically homogeneous throughout the mediolateral axis of the cerebellum.

To see whether the administration of HU alters the neurogenetic timetables of PCs and DCN neurons, different sets of saline and treated rats were examined. Our results have revealed that, in both macroneurons, the entire span of neurogenesis, including its pattern of peaks and valleys, and the peak production were different between rats administered with saline or HU. This occurs in each of the analyzed cortical compartments and in deep nuclei. For example, in saline-injected rats, the neurogenesis of PCs and DCN neurons occurred between days E12 to E15, with a PC production peak at day E14 in each cortical compartment, as well as in each deep nucleus. In the HU-treated group, on the other hand, developmental timetables extended from day E15 until day E19 for PCs and DCN neurons, with a peak generation at E17 in each compartment of the cerebellar cortex as well as in each deep nucleus. These results indicate that day E12 is a time of high susceptibility to insult. Moreover, it is shown here that the temporal sequence of PCs and DCN neuron production throughout the cerebellum was disturbed, suggesting that the neuroblasts that give rise to these macroneurons are highly susceptible to HU.

An important aspect during the developmental injury of the cerebellar cortex and the deep nuclei is ascertaining whether the spatial location of PCs and DCN neurons was modified due to HU exposure. Our results revealed two gradients of cytogenesis for these macroneurons in saline rats. The first gradient is related to the PCs. There is a tendency for the PCs located in the region of the vermis to be generated after those found in the lateral hemispheres. In the second cytogenetic pattern, the DCN neurons located in the fastigial nucleus arise earlier than those from the dentate nucleus. In HU-treated rats on the other hand, the above-mentioned neurogenetic gradients were modified, indicating that the arrangement of cortico-nuclear connections in the cerebellum may be altered.

Two observations emerge from these experiments: HU-exposure decreases the size of the cerebellar cortex and deep nuclei and secondly that this cytotoxic agent compromises the survival of PCs and DCN neurons, disturbing the times of neuron origin and the neurogenetic gradients of these macroneurons.

## 8. Perinatal Effects of HU Exposure: Short-Survival Experiments

Early experiments have shown that HU exposure in perinatal life causes cell depletion in the EGL and malpositioning of GCs [10,11]. Despite these data, the effects of HU on the early postnatal development of the cerebellar cortex have not been completely elucidated. In this section, we characterize the type of cell death of EGL neuroblasts induced by HU exposure in early postnatal life. In addition to this, we also analyze the morphological and ultrastructural changes of Bergamann glial cells, and the microglial response after HU treatment. These data are taken from previous published papers [61,62].

Groups of rats were injected with a single injection of saline or HU (2mg/g b.w.) on day P9 and sacrificed at different survival times, ranging from 6 to 24 h, and 48 and 72 h after saline or HU exposure. Following paraffin embedding and tissue processing, cell death was analyzed in the EGL during treatment, and its kinetic pattern was established. The dying neuroblasts exhibited characteristic apoptotic features in TUNEL labeling cerebellar sections (Figure 2a). To confirm the morphological features of cell death, light and electron microscopy was used.

Our results showed that when plastic thin sections were studied with light microscopy, dying cell profiles were found (Figure 2b). The ultrastructural analysis showed the presence of neuroblasts at different stages of apoptosis (Figure 2c–f). The earliest signs of this type of cell death were the condensation of the chromatin and its segregation against the inner nuclear envelop. The mid-to-late apoptotic stage was characterized by the presence of several spherical and electron-dense nuclear fragments. Finally, numerous clusters of apoptotic bodies were visualized. From these experiments, it was deduced that HU administration activates apoptotic cellular events, resulting in a substantial depletion of cells. Activation of apoptosis has also been reported in the EGL after hyperoxia [63], X-ray exposure [64] and the administration of some treatments, such as lead [65], ethanol [66] and platinum compounds [50]. In this context, the experimental model of HU-induced apoptosis reported here could provide a good system to study the apoptotic mechanisms in the developing cerebellum.

We also analyzed the effect of HU on the viability and morphology of the Bergmann glial cells. After some immunohistochemical procedures, it was observed that HU decreases the number of Bergmann glial cells with respect to saline rats. Despite that, the typical palisade organization of this unipolar astrocyte was preserved. Moreover, our results also show the overexpression of the cytoskeletal protein vimentin and the formation of thicker immunoreactive glial processes, including the end-feet at the pial surface, in those surviving Bergmann glial cells [61]. The damage of radial glia may cause an alteration of the migratory pattern of GCs, Purkinje dendrite differentiation and the development of synaptic ensheathment.

Our electron microscope analysis reveals that Bergmann glial processes present phagosomes containing apoptotic bodies and cell debris. This suggests that the surviving Bergmann glial cells can serve as facultative phagocytes. Moreover, our ultrastructural images indicate that dying cells at different stages of apoptosis were covered by laminar processes of Bergmann glia (Figure 2g). We propose that this tight relation may be isolating the degenerating cells in closed compartments in order to protect the undamaged neuroblasts against apoptosis. Several studies have indicated that programmed cell death via apoptosis may be a communal event. In this context, there are evidences indicating that during the development of *Drosophila melanogaster*, dying cells produce Eiger, a tumour necrosis factor ortholog to trigger apoptotic events in normal cells through the activation of the c-Jun N-terminal kinase pathway. In mice, on the other hand, there are data showing that during hair follicle development, several follicle cells undergo apoptosis, which is dependent on the TNF- α secretion by apoptotic cells [67]. From these results, we hypothesized that the tight association between Bergmann glial cells and degenerating EGL neuroblasts is to protect healthy cells from apoptotic cell signalling molecules produced by dying cells.

Microglia reside in the central nervous system, where they function as immune cells. Previous studies have indicated that these cells become activated in response to injury in order to maintain brain microenvironment homeostasis. Microglia activation plays an important role in the phagocytosis of dead cells or cellular debris [68]. As our studies have revealed that HU-exposure induces apoptosis in the developing cerebellum, we examined whether the damaged of EGL cells can lead to the activation of microglial cells. Tomato lectin histochemistry and transmission electron microscopy revealed that ameboid microglial cells participate in the phagocytosis of injured neuroblasts in regions of the EGL with extensive cell death. Moreover, electron micrographs show activated microglia adjacent to injured EGL cells, and containing apoptotic figures and cellular debris (Figure 2h). This suggests that the signals produced by apoptotic cells may modify the dynamic behavior of microglia and trigger the recruitment and activation of glial cells to remove injured cells and repair the brain parenchyma.

In order to obtain more information about the effects of HU exposure on the development of the EGL, groups of rats were administered with this pharmacological agent at days P5, P10 or P15. In each of these ages, animals were sacrificed and examined at appropriate times, ranging from 6 to 48 h after treatment administration. Studies were done in the cerebellar cortex lobe following the quadrupartite lobular division. Our results demonstrate that the vulnerability of EGL neuroblasts and Bergmann glia, just as the microglial activation, depends on the analyzed postnatal day, vermal lobe and survival time after drug exposure. Evidence presented here denotes that the most important alterations occurred at P10, indicating that this is an age of high vulnerability to injury. It is also indicated here that EGL cells located in the anterior and central lobes are the most susceptible to the action of the HU. Moreover, the time span from 6 to 24 h is a period of time of high vulnerability to this antimetabolite. We suggest that these temporal and zonal variations in sensibility of EGL neuroblasts to HU could be related to differences in the time of production of GCs during the early postnatal development. The anterior and central lobes contain, in general, late-generated GCs [36] and these regions are more vulnerable to HU administration. On the other hand, the posterior and inferior lobes have, in general, early-born GCs [36] and these lobes are less vulnerable to HU exposure. 

## 9. Perinatal Effects of HU Exposure: Long-Survival Experiments

The structural and functional organization of the cerebellum is the end product of a complex developmental process. The cerebellar neurons, including PCs and GCs, are produced in overlapping waves and migrate to their final locations to achieve the final cerebellar structure and the organization of its circuits [36]. The disruption of this process may produce alterations in the arrangement of the cortical neurons. To our knowledge, no attempts have been made to determine the long-term effects of HU administration on the development and cytoarchitectonics of the cerebellar cortex.

In this section, we show that a single administration of HU in the early postnatal life alters the spatial location of PCs and GCs and the arrangement of the PC dendritic tree. The data presented here were taken from a previously published paper [61]. In the study, groups of P9 rats were injected intraperitoneally with a single injection of saline or HU (2 mg/g b.w.). Then at P12, they were administered with BrdU (50 mg/kg b.w.; i.p.). The animals were sacrificed at P45. Our results revealed that, in relation to saline, the HU condition disturbs the developmental program of the cerebellum, resulting in anomalies in the cortical cytoarchitecture. These included:

(1) The ectopic placement of PCs and abnormalities in their morphology (Figure 3).

Examination of calbindin D-28k immunostained sections indicated that these macroneurons were piled 2–3 cell-thick. Some of these cells were found in the granular layer (GL). Moreover, in many PCs the following morphological abnormalities in the dendritic arborization were observed: (A) The ascending dendritic tree bifurcated in a T-shape manner, (B) the primary dendrite ascended obliquely to the pial surface, (C) the primary dendrite was oriented toward the GL, and (D) the neuronal body and dendritic tree were directed toward the white matter. Interestingly, the dendritic alterations of PCs were mainly encountered in regions where GC ectopia was presented.

(2) The malposition of GCs. In all cortical lobes, a large number of ectopic GCs forming a supernumerary layer were found in the ML. The position of the ectopic GCs presented four patterns: (A) Small clusters of cells located near the cerebellar surface. These were observed in the lobules I and X as well as at the bottom of the fissures prima and secunda. (B) Arrangement in a monolayer just beneath the pia mater. This was seen in the lobules II and III. (C) Imprecise aggregation in a thin strip oriented in parallel to the pial surface. This band is usually found in the upper ML. This pattern was found in the lobules IV, V and IX. (D) Occupying the middle-to-lower part of the ML (lobules VI to VIII) and parallel to the pial surface.

All these data indicate that the administration of HU in the early postnatal life causes anomalies in the cortical cytoarchitecture in the adulthood.

## 10. Conclusions

The current review describes and clarifies the effect of HU treatment on the development of the rat cerebellum. Although HU is reported as a teratogen for rodents, the doses administered to experimental animals is several times higher than the therapeutic ones., Because of this dose difference, extrapolation from rats to humans may be doubtful. Despite that, we have shown here that the immature cerebellum is highly vulnerable to HU exposure. A single injection of this agent in utero triggers apoptotic cell events in the cerebellar neuroepithelium and alters the developmental timetables and the neurogenetic gradients of PCs and DCN neurons. Moreover, when administered in the perinatal life, HU produces the apoptotic elimination of EGL neuroblasts, and a reactive response of the Bergmann glial cells and microglia. Our results also reveal that HU exposure decreases the size of the cerebellum and induces alterations of the cortical cytoarchitectonics, including dendritic alterations of PCs, which occur in parallel with the demise of GCs and ectopic location of these neurons. In the early postnatal period, the effect of HU exposure depends on the analyzed postnatal day, vermal lobe and survival time after drug exposure. As cerebellar dysfunctions contribute to neuropathological conditions such as autism, mental illness and behavioral disorders [69,70], we advise that extreme caution should be taken when administering HU to gestating women or infants as the effects of this agent on the cerebellum might persist throughout their offspring’s lives. More studies are needed to determine with certainty whether HU administration is safe for human embryos and young children.

## Figures and Tables

**Figure 1 ijms-19-03986-f001:**
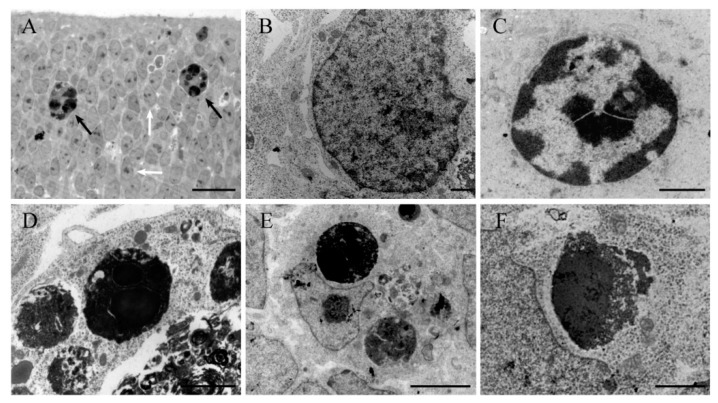
(**A**) Light microscope view of a thin plastic section from a rat cerebellum exposed to hydroxyurea in the prenatal life. The section presents among healthy neuroblasts (white arrows) and apoptotic profiles (black arrows) with dark spherical balls distributed within the cell. (**B**–**F**) Ultrastructural morphology of healthy (**B**) and apoptotic neurons (**C**–**F**) in the rat neuroepithelium following HU administration. (**C**) Early stage of apoptosis showing nuclear chromatin at the margin of the nucleus. (**D**,**E**) Late apoptotic stage showing typical clusters of apoptotic bodies. (**F**) Breakup of an apoptotic body and release of its contents into the cytoplasm. Scale bar: 20 µm (A), 1 µm (B,C), 2 µm (D,F), 5 µm (E).

**Figure 2 ijms-19-03986-f002:**
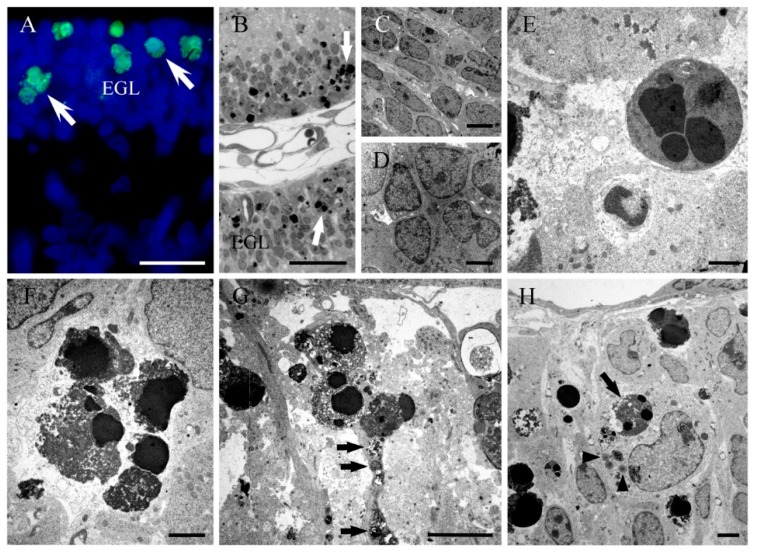
(**A**) TUNEL positive cells in the cerebellar external granular layer (white arrows) from a rat administered with hydroxyurea in perinatal life. (**B**) Thin plastic section (0.5 µm) from a rat cerebellum following hydroxyurea treatment. The healthy neuroblasts have a homogeneous nucleoplasm contained within an intact nuclear membrane. The apoptotic cells (white arrows) present two or more spherical balls within the cell. (**C**–**H**) Electron micrographs of healthy (**C**,**D**) and apoptotic neurons (**E**,**F**) in the external granular layer after hydroxyurea exposure, showing characteristic clusters of apoptotic bodies. (**G**) Bergmann glial processes containing electron-dense phagosomes (black arrow) in a hydroxyurea-treated animal. (**H**) Electron micrograph of an ameboid microglial cell engulfing an apoptotic body (black arrow) and the presence of several lipid droplets (head arrow). EGL: External granular layer. Scale bar: 20 µm (A), 25 µm (B), 5 µm (C,G), 2 µm (D,F,H), 1 µm (E).

**Figure 3 ijms-19-03986-f003:**
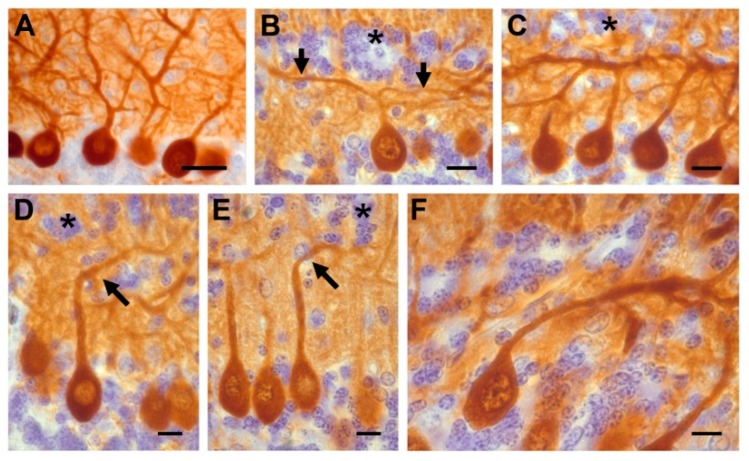
(**A**) Calbindin-reactive Purkinje cells in saline rats whose dendrites grew in the normal, upward direction. (**B**–**F**) Hydroxyurea-treated cerebella showing different patterns of dendritic arborization. Asterisks show ectopic interneurons and black arrows indicate the altered dendritic tree. Scale bar: 30 µm (A); 10 µm (B–F).

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
