# Peer review of "An Animal Model for Assessing the Effects of Hydroxyurea Exposure Suggests That the Administration of This Agent to Pregnant Women and Young Infants May Not Be as Safe as We Thought"

_ijms, 2018, doi:10.3390/ijms19123986_

Reviewer 1 Report

The authors have done an excellent job comparing their findings with previous reports. The manuscirpt is extremely clear for the reader and well written. This review will be important for future use of Hydroxyurea administration in pregnant women and young infants.

Author Response

Thank you for your review of the present manuscript. Your comments have been very valuable to us.

Reviewer 2 Report

In the review by Rodriguez-Vazquez and Marti the authors describe the results of previous experimental studies by the same authors showing that treatments with hydroxyurea (HU) during embryonic and perinatal stages in rats cause significant, stage-dependent alterations of cerebellar development. The manuscript is clearly written and the summary of their previous work provided by the authors in this review will be useful to researchers working on cerebellar development or clinicians investigating the negative-side effects of HU when used for therapeutic purposes. Listed below are some minor issues that the authors could deal with before publication:

1) It would be useful to include a description of the doses of HU used for the clinical treatment of human patients. Are they comparable to the doses that affect brain development in rats?

2) Pag. 3, rows 129-131: the sentence is not clear and might be changed to: "As the cerebellar neuroepithelium presents a cohort of asynchronous cycling cells in S-phase, neurotoxic factors that disrupt the proliferative activity of neural progenitors, such as HU, may cause cell death in the cerebellum."

3) Pag. 4, rows 140-141: the authors claim that the neural progenitors that resume mitotic activity following HU treatments are cells that have repaired the damage caused by HU. Are actual experimental evidences for this repair? If so, the relevant papers should be quoted. Otherwise, is it possible that the cells that resume mitotic activity are simply those that, for whatever reason, have been spared by HU treatments?

4) Pag. 8, rows 308-310: it may be better to slightly change the text to: "...groups of rats were administered with this pharmacological agent at P5, P10 or P15. In each of these ages, animals were sacrificed and examined at appropriate times ranging from 6 to 48h after treatment administration".

5) Pag. 8, rows 317-320: it might be useful to clarify that the differential susceptibility of the anterior/central and the posterior/inferior lobes of the cerebellum to HU treatments are relative to HU treatments performed during postnatal stages.

Author Response

Thank you very much for your review of the present manuscript. Your constructive suggestions have been very valuable to us. The following points have been modified in accordance with your recommendations. We think that we have now responded clearly to the reviewer’s comments.

The reviewer wrote (in bold, cursive writing):

1) It would be useful to include a description of the doses of HU used for the clinical treatment of human patients. Are they comparable to the doses that affect brain development in rats?

We are grateful to you for this comment. The doses of HU used for the therapeutic treatment of human patients and the doses that affect the normal development of the rat cerebellum is describe See:

- Pages 2-3, rows 86-93, section entitled: 4. Teratogenic effects of hydroxyurea.

- Page 10, rows 392 to 394, section entitled: 10. Conclusions.

2) Pag. 3, rows 129-131: the sentence is not clear and might be changed to: "As the cerebellar neuroepithelium presents a cohort of asynchronous cycling cells in S-phase, neurotoxic factors that disrupt the proliferative activity of neural progenitors, such as HU, may cause cell death in the cerebellum."

We are grateful to you for this comment and accept that you are right. We apologize for the lack of clarity in the first version. Following your suggestion, the sentence has been modified (see page 3, rows 137-138, section entitled: 6. Embryonic effects of HU exposure: short-survival experiments.

3) Pag. 4, rows 140-141: the authors claim that the neural progenitors that resume mitotic activity following HU treatments are cells that have repaired the damage caused by HU. Are actual experimental evidences for this repair? If so, the relevant papers should be quoted. Otherwise, is it possible that the cells that resume mitotic activity are simply those that, for whatever reason, have been spared by HU treatments?

We are grateful to you for this comment and accept that you are right. This paragraph has been modified following the suggestions of the reviewer (see page 4, rows 148-150, section entitled: 6. Embryonic effects of HU exposure: short-survival experiments). 

4) Pag. 8, rows 308-310: it may be better to slightly change the text to: "...groups of rats were administered with this pharmacological agent at P5, P10 or P15. In each of these ages, animals were sacrificed and examined at appropriate times ranging from 6 to 48h after treatment administration".

We are grateful to you for this comment. We apologize for the lack of clarity in the first version. Following your suggestion, the sentence has been modified (see page 8, rows 330-332, section entitled: 8. Perinatal effects of HU exposure: short survival experiments).  

5) Pag. 8, rows 317-320: it might be useful to clarify that the differential susceptibility of the anterior/central and the posterior/inferior lobes of the cerebellum to HU treatments are relative to HU treatments performed during postnatal stages.

We are grateful to you for this comment. We apologize for the lack of clarity in the first version. Following your suggestion, the sentence has been modified (see page 8, rows 339-345, section entitled: 8. Perinatal effects of HU exposure: short survival experiments).